# Douglas Fir Growth Is Constrained by Drought: Delineating the Climatic Limits of Timber Species under Seasonally Dry Conditions

Antonio Gazol [1], Cristina Valeriano [1], Alejandro Cantero [2], Marta Vergarechea [3] and Jesús Julio Camarero [1,*]

1    Instituto Pirenaico de Ecología (IPE-CSIC), Avda. Montañana 1005, 50059 Zaragoza, Spain
2    HAZI Fundazioa, 01192 Arkaute, Spain
3    Norwegian Institute for Bioeconomy Research (NIBIO), Division of Forest and Forest Resources, National Forest Inventory, Høgskoleveien 8, 1433 Ås, Norway
*    Correspondence: jjcamarero@ipe.csic.es; Tel.: +34-976-369-393

**Abstract:** There is debate on which tree species can sustain forest ecosystem services in a drier and warmer future. In Europe, the use of non-native timber species, such as Douglas fir (*Pseudotsuga menziesii* [Mirb.] Franco), is suggested as a solution to mitigate climate change impacts because of their high growth resilience to drought. However, the biogeographical, climatic and ecological limits for widely planted timber species still need to be defined. Here, we study the growth response to climate variables and drought of four Douglas fir plantations in northern Spain subjected to contrasting climate conditions. Further, we measure wood density in one of the sites to obtain a better understanding of growth responses to climate. Correlative analyses and simulations based on the Vaganov–Shaskin process-based model confirm that growth of Douglas fir is constrained by warm and dry conditions during summer and early autumn, particularly in the driest study site. Minimum wood density increased in response to dry spring conditions. Therefore, planting Douglas fir in sites with a marked summer drought will result in reduced growth but a dense earlywood. Stands inhabiting dry sites are vulnerable to late-summer drought stress and can act as "sentinel plantations", delineating the tolerance climate limits of timber species.

**Keywords:** basal area increment; dendroecology; minimum wood density; *Pseudotsuga menziesii*; plantations; radial growth; Vaganov–Shaskin growth model

## 1. Introduction

Climate change is expected to cause air temperatures to continue to increase and modify seasonal rainfall patterns in drought-prone regions, such as the Mediterranean Basin [1]. Thus, we can expect that climate-sensitive ecosystems, such as forests, will face major changes in structure, composition and functioning in the coming decades [2]. Under more arid climate scenarios, models forecast an upward and poleward migration of Mediterranean tree species towards wetter and cooler areas [3]. In consequence, it is expected that future Mediterranean forests will be dominated by more drought-tolerant and less productive tree species [2]. It is, then, not surprising that forest managers, stakeholders and researchers are searching for tree species capable of resisting climate change and showing high growth resilience to more frequent and severe droughts [4,5]. This search is particularly relevant in the Mediterranean Basin because it is a climate change hotspot [6]. Interestingly, some areas from this region, such as north-eastern Spain, where several temperate tree species find their southern distribution limit (e.g., Ref. [7]), can serve as a basis for what could happen in less dry, temperate regions across central and western Europe, particularly in terms of growth responses to drought stress [5].

Over the last few decades, it is becoming increasingly apparent that drought is one of the most important global drivers of tree growth [8]. Indeed, drought stress is recurrently cited as a major cause of forest dieback and elevated tree mortality [9]. For instance, in central Europe, the intense and severe droughts and heatwaves observed in the years 2003 and 2018 led to the occurrence of widespread drought-induced dieback and tree mortality of many species [10]. These impacts, together with the abovementioned expected responses of European forests to ongoing climate change, trigger debates on which tree species should compose future European forests (e.g., Ref. [11]). Potential solutions include assisted migrations and selection of more resilient varieties of native trees (e.g., Ref. [5]), but also use of planted timber species with the potential to cope with the forecasted warmer and drier climate conditions [12]. Therefore, it is urgent to understand how those non-native tree species widely planted in Europe respond to drought to advance in the understanding of how they will thrive under more arid conditions.

Douglas fir (*Pseudotsuga menziesii* [Mirb.] Franco) is among the most widely planted non-native timber species in Europe, with plantations growing under cool–wet conditions in Scandinavia, central and Western Europe or under seasonally dry conditions in Spain, Italy and Turkey [13]. This conifer belongs to the Pinaceae family, and it is native to western North America, where two varieties exist: *P. menziesii* var. *menziesii* (coastal variety), which is distributed from central British Columbia south along the Pacific coast ranges into central California, and *P. menziesii* var. *glauca* (interior variety), which inhabits the Rocky Mountains from the USA to northern Mexico. According to a recent review, most European plantations correspond to the coastal variety [4]. In western and central Europe, Douglas fir has been widely planted because of its rapid height growth, efficient shading of competing trees, good drought tolerance, high nutrient- and water-use efficiency and timber quality, all factors that make this tree species attractive to foresters [14–16]. Therefore, it is being considered as a promising alternative to other native tree species because it could have greater resilience capacity against drought and biotic stressors (e.g., bark beetles) than native conifers, such as Norway spruce or silver fir [14]. It has been suggested that Douglas fir may become a relevant timber species of future European forests and could have strong potential to mitigate climate change impacts through carbon sequestration [4,15,16]. Despite some of the benefits of the species relying only on its capacity to recover after a water shortage, it is also vulnerable to severe droughts, particularly in sites with less fertile soils [17]. In Spain, Douglas fir covers 45,000 ha, mostly located in the northern Cantabrian coast and nearby ranges, such as the north-western Iberian System and the western Pyrenees, where some of the driest sites of the species across Europe are located [18]. By the end of the 19th century, single individuals of Douglas fir were planted in Vizcaya, northern Spain [18], but plantation of the species started to become widespread in regions of northern Spain (País Vasco, La Rioja, Navarra) from the 1920s onwards.

Several studies have evaluated how Douglas fir growth responds to drought both in North America [19] and also in Europe [16,20] to assess the species' vulnerability to water shortage. For instance, widespread Douglas fir growth decline was found in south-western North America because of drought [19]. Others reported that growth of Douglas fir was reduced by water deficit, particularly in drier sites [21], with trees from provenances inhabiting more arid regions being more resilient against drought [22]. Moreover, an increase in growth sensitivity to drought over the last few decades has been reported in western and central Europe [23]. However, we lack a reliable definition of the climate limits of the species based on long-term growth data to better assess how Douglas fir responds to droughts of different intensity and duration depending on site conditions. In this study, we fill this knowledge gap by analyzing the radial growth responses to climate variability in Douglas fir plantations subjected to contrasting climatic conditions, including xeric sites, which may represent the climatic limits of tolerance of the species. We used those arid sites and calculated climate–growth relationships as proxies of the limits of drought tolerance in this major timber species.

Here, we argue that comparing plantations subjected to different drought intensity and duration will help to better delineate the climate limits for this timber species in Europe. To this end, we sampled adult Douglas fir trees in four sites subjected to contrasting climate conditions located in northern Spain, near the southernmost distribution limit of Douglas fir plantations in Europe [18]. Further, we combined tree-ring-width series with wood density data in one of the populations to better understand how the wood formation and radial growth of this species respond to drought. We used several statistical models and compared them with a process-based growth model to assess the vulnerability to low soil moisture of Douglas fir plantations following similar analyses in natural stands [24]. We hypothesize that: (i) Douglas fir growth will be constrained by drought and heat in the driest site (Cornago), reflecting sub-optimal growth conditions, whereas growth rates will be higher in the wetter sites; and (ii) minimum wood density will reflect Douglas fir responses to drought during the early to middle portion of the growing season. Specifically, we expect Douglas fir would form dense earlywood, leading to high minimum wood density values due to formation of tracheids of narrow lumen in response to the water deficit during the early to middle portion of the growing season [25,26]. According to wood anatomical studies, when soil water availability is reduced during the early to middle portion of the growing season, conifers form conduits of narrow lumen [27].

In this study, we used as a reference site the driest Cornago site because it is located in a region with Mediterranean and continental climatic influences, where a longer summer drought due to warmer climate conditions can reduce its growth substantially and threaten plantation viability. We argue that plantations inhabiting extreme, dry sites are vulnerable to drought stress and can be used as "sentinel plantations" of forecasted aridification. Our dendroecological framework allows defining the climatic limits of tolerance of Douglas fir, a major timber species in Europe, under seasonally dry conditions.

## 2. Materials and Methods

### 2.1. Study Sites

We studied four Douglas fir plantations located in northern Spain and subjected to different climate conditions (Figure 1 and Figure S1). The characteristics of the four sites are provided in Table 1. The climate in Opakua and Beamaburi sites is mainly temperate, with cold winters and mild summers, whereas climate is warmer and drier in Cornago and Villaverde de la Rioja (hereafter Villaverde) sites, particularly in the Cornago site, due to a stronger influence of Mediterranean climate.

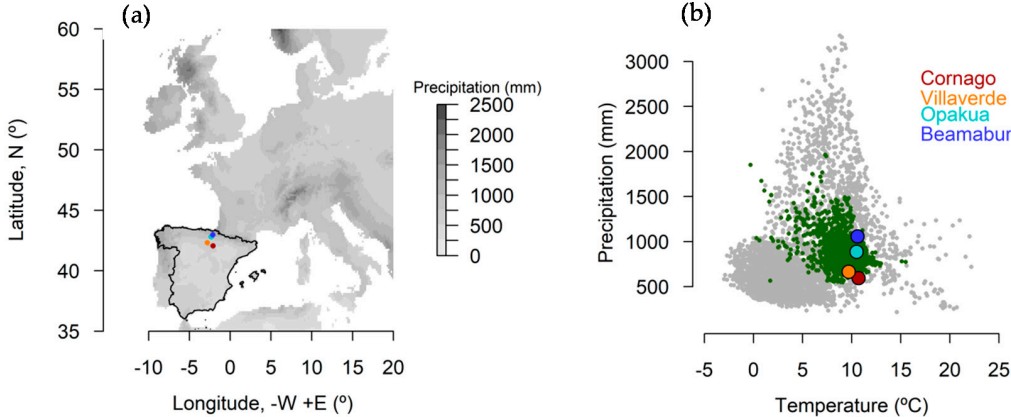

**Figure 1.** (**a**) Location of the four study sites in northern Spain and (**b**) climate niche space of Douglas fir in North America (grey points), Europe (green points) and the four studied sites (color points) based on annual climate data (mean temperature, total precipitation). Note that the Cornago and Villaverde sites are subjected to drier conditions than the Opakua and Beamaburu sites.

**Table 1.** Characteristics of the sampled sites. Climate data are annual.

| Site | Latitude N | Longitude W | Elevation (m a.s.l.) | Temperature (°C) | Precipitation (mm) | Main Lithology |
|------|-----------|------------|---------------------|-----------------|-------------------|----------------|
| Villaverde | 42.321 | 2.812 | 886 | 9.8 | 640 | Shale |
| Opakua | 42.818 | 2.323 | 1023 | 9.9 | 990 | Conglomerates |
| Beamaburu | 42.985 | 2.149 | 880 | 12.0 | 1110 | Limestone |
| Cornago | 42.065 | 2.107 | 1023 | 12.1 | 570 | Shales |

The mean (±SD) diameter at 1.3 m and total height of sampled trees varied from 25.6 ± 3.1 cm and 11.8 ± 1.0 m in Cornago to 39.6 ± 3.3 cm and 36.1 ± 4.8 m in Beamaburu. The mean distance between trees ranged between 2.8 ± 0.1 m in Cornago and 3.9 ± 0.4 m in Villaverde. We did not observe stumps or pruned branches in the sampled sites, which could suggest past thinning or other management activities.

We also characterized the natural (North America) and planted (Europe) climate niches of Douglas fir using distribution maps across North America and Europe created by combining different sources of information [28,29]. Polygons or raster layers from the distribution for each species were converted into points and the climate conditions were obtained from each point. Total annual precipitation and mean annual temperature were downloaded from the WorldClim database [30] at 10′ resolution. The climate space for each species was described by the climate conditions in each point (see Figure 1).

### 2.2. Field Sampling

In each site, between 12 and 20 trees were sampled for dendrochronological purposes. Dominant or co-dominant trees were selected within the stands and avoiding trees separated less than 5 m. The diameter at breast height (dbh) was measured at 1.3 m for each sampled tree using tapes, and two 5-mm-thick cores were extracted in perpendicular direction using Pressler increment borers (Häglof, Sweden).

### 2.3. Tree-Ring Width Data

The cores were prepared following standard dendrochronological procedures [31]. They were air-dried, glued into wooden supports and their surface was sanded to enhance the visibility of tree-ring borders. These processed cores were visually cross-dated. Then, we measured the tree-ring widths of each sample up to the precision of 0.01 mm under a binocular using a Lintab-TSAP measuring device (Rinntech, Heidelberg, Germany). We validated the visual cross dating by calculating moving correlations between the individual series and the mean series of each species. Cross-dated series were used to study Douglas fir growth trajectories and response to drought at the site and tree levels.

### 2.4. Climate and Drought Data

For each site, we downloaded elevation-corrected mean temperature and total precipitation data at daily and monthly scales from the CRU database v. 4.05 (data available at https://crudata.uea.ac.uk/cru/data/hrg/, last accessed on 18 August 2022; [32]). Temperature is increasing in all study sites (Figure S2).

To characterize drought intensity in each site, we used the Standardized Precipitation Evapotranspiration Index (SPEI), which is a proxy of soil moisture [33]. This normalized drought index is quantified based on the difference between precipitation and potential evapotranspiration. SPEI data from the studied sites was downloaded from the Global Drought Monitor webpage (http://spei.csic.es/database.html, last accessed on 19 August 2022; [33]). In both cases, climate data were gridded at 0.5° resolution. The study area was impacted by the severe 1994, 2001, 2005 and 2012 droughts [7].

### 2.5. Building Tree-Ring Width Chronologies

To study climate–growth relationship at the site level, we constructed mean site detrended ring-width series or chronologies. Each individual ring-width series was detrended

with a cubic smoothing spline with a length of 20 years and a 0.5 response cut-off to remove biological trends due to tree aging and increase in size. Indices were calculated by diving observed by fitted values. Note that the same spline length was used across sites to enable comparison of the chronologies and their response to climate [34]. In addition, temporal autocorrelation was removed by pre-whitening each series using ARIMA models. Finally, pre-whitened series were averaged using bi-weight robust averages to obtain residual chronologies. All chronologies were trimmed to cover the period 1990–2014, which is a period of sufficient replication common to all sites. We calculated the mean correlation between individual series (Rbar) and the Expressed Population Signal (EPS) in order to estimate within-species growth coherence [30].

### 2.6. Climate–Growth Relationships at Tree Level

To study growth trajectories and responses to climate at the tree level, tree-ring widths measures were converted in basal area increments (BAI). We used the equation:

$$BAI = \pi \left( r_t^2 - r_{t-1}^2 \right) \tag{1}$$

which assumes a circular shape of stems and where $r_t$ and $r_{t-1}$ are the tree radius in the year of tree-ring formation ($t$) and the year before tree-ring formation ($t - 1$), respectively. We transformed tree-ring width measures in BAI because it adequately reflects growth trajectories in the early stages of tree life and retains high-frequency variations. BAI was calculated for each individual series from the outside inwards. After that, the two measures for each tree were averaged.

### 2.7. Wood Density Data

In the Beamaburu site, ring-width series from 16 trees were analyzed using an ITRAX X-ray microdensitometer (Cox Analytical Systems, Gothenburg, Sweden) at the CETEMAS (Centro Tecnológico y Forestal de la Madera, Carbayin, Spain). Before being analyzed, wood samples were air-dried to have a moisture content of 12%. Each sample was scanned using the ITRAX (with standard X-ray intensity of 30 kV and 35 mA and exposure time of 20 ms), recording X-ray images with a geometrical resolution of 40 measurements per millimeter (cf. Ref. [35]). The X-ray images were further processed to create intra-ring density profiles for each wood specimen. We paid particular attention to the minimum wood density (MND) of the ring, which is similar to the earlywood density, and it has been understudied [26] despite showing strong responses to spring drought [36,37]. We also considered the maximum (MXD) and ring (RGD) density values. We expect the MXD to respond to changes in mid to late growing season temperatures affecting cell-wall thickening of tracheids [25]. To account for tree age effects on wood density, we calculated detrended (age-corrected) wood density residual indices (Figure S3), which were obtained as in the case of ring-width indices but by subtracting observed from fitted values.

### 2.8. Relationships between Climate and Tree-Ring Width or Density Series

Correlation analyses were used to test for the relationship between ring-width indices or density indices and average monthly maximum temperature and drought severity or duration (SPEI) for the period 1990–2014. Bootstrapped correlation analyses were performed, and their significance was assessed by resampling 1000 times [38]. Analyses were performed at the site level using the site residual chronologies and the maximum temperature and 3-month-long SPEI from the month of September in the year before growth to the month of September in the year of growth. Correlation analyses with the same procedure were used to test for the relationships between wood density (MND, MXD and RGD) and climate variables in Beamaburu.

### 2.9. Process-Based Growth Model

To evaluate how growth variability depended on climate, we simulated the intra-annual growth patterns (RWI, ring-width indices) of the four Douglas fir sites with the

Vaganov–Shaskin (VS) process-based model based on its version 1.37 [24,39]. As input variables, daily climate data (mean temperature and total precipitation) and residual series of ring-width indices for the best-replicated period in each site were used. To compare observed and simulated growth series [40], Pearson correlations, the synchronicity index (Gleichläufigkeit statistic, Glk) and root-mean-square errors (RMSE) were calculated (Table S2). The model simulates integral growth rates (Gr) and partial growth rates due to temperature (GrT) and soil moisture limitations (GrM). We also analyzed growth sensitivity to climate variables through time based on the model simulations [41].

### 2.10. Statistical Analyses

To compare mean tree-ring values between sites, we used *t*-tests. We used linear mixed-effect models (LMM; Ref. [42]) to study growth trajectories and response to climate in each site at the tree level. The model was of the form:

$$Y = f\ (Xst) + us + vt + est \tag{2}$$

where $Y$ is the BAI of each individual $s$ in the year $t$, $f(X)$ is the set of fixed effects, $us$ represents the tree nested within stands random effects, $vt$ is a normally distributed random effect for calendar year $t$ (year effect) and $est$ is the normally distributed residual for tree $s$ at year $t$. We used this random structure for the model following Ref. [43] to study the temporal variation in BAI while accounting for the fixed effects together with unspecified tree-, and year-level factors.

As fixed factors, we included the age trend, the SPEI and the site factor (i.e., four levels). The age trend was represented by the log-transformed (log(x+1)) age of each tree $s$ at year $t$. To represent drought conditions, we selected the 3-month-long SPEI for the month of August and the mean maximum temperature of August. We expect that these climate conditions capture the dependency of the growth of this species to summer conditions based on previous studies [20]. We included potential interactions between both variables and site. The analyses were carried out for the period common to the four sites (1990–2014). Fixed factors were standardized to have zero mean and unit variance prior to the analyses and centered in the case of the SPEI and temperature. Model selection was based on minimizing the Akaike Information Criterion (AIC) [44]. All potential models combining the variables listed above were created and the one with the lowest AIC was selected. The model was evaluated by graphically inspecting the residuals [45]. For the selected model, we calculated the conditional ($R^2c$; variance explained by fixed plus random effects) and marginal $R^2$ ($R^2m$; variance explained by fixed effects) coefficients of determination according to Ref. [46].

The analyses were performed in the R environment for statistical computing [47] using Rstudio [48]. We used the package dplR [49,50] to manage tree-ring-width series, detrend them and calculate BAI. The package lme4 [51] was employed to create linear mixed-effect models, the package MuMIn was used to perform AIC model selection [52] and the package emmeans [53] was used to perform trends and factor comparisons using estimated marginal means. Finally, the effects package was used to visualize regression graphs [54].

## 3. Results

### 3.1. Growth Variability between Sites and Years

Douglas fir radial growth varied considerably between sites (Table 2). As expected, trees formed the widest rings in the wettest Beamaburu site. Conversely, the narrower rings were found in the driest Cornago site, where the trees were a bit younger than in the rest of the sites (Table 2), and their BAI was clearly lower (Figure 2). First-order autocorrelation was lower in Opakua than in the rest of the sites, probably due to the strong impact of the 2005 drought (Figure 2). The inter-series correlation and the EPS were similar across sites, excluding the relatively low values found in Beamaburu. We considered the period 1990–2014 to calculate correlations between climate and ring-width indices or density

indices because it was the best replicated. All the sites showed EPS > 0.85 during the whole period.

**Table 2.** Dendroecological statistics for Douglas fir calculated for the common period 1990–2014. Different letters indicate significantly ($p < 0.05$) different values between sites according to *t*-tests. Values are means $\pm$ SD.

| Site | No. Cores (No. Trees) | Period | Autocorrelation | Ring Width (mm) | Rbar | EPS |
|---|---|---|---|---|---|---|
| Villaverde | 39 (20) | 1976–2014 | 0.53 ± 0.17 | 4.49 ± 1.71 b | 0.67 | 0.89 |
| Opakua | 25 (13) | 1982–2021 | 0.57 ± 0.21 | 4.58 ± 1.31 b | 0.52 | 0.87 |
| Beamaburu | 15 (15) | 1977–2017 | 0.60 ± 0.26 | 5.20 ± 1.80 b | 0.47 | 0.85 |
| Cornago | 24 (12) | 1990–2021 | 0.64 ± 0.25 | 3.57 ± 1.25 a | 0.74 | 0.90 |

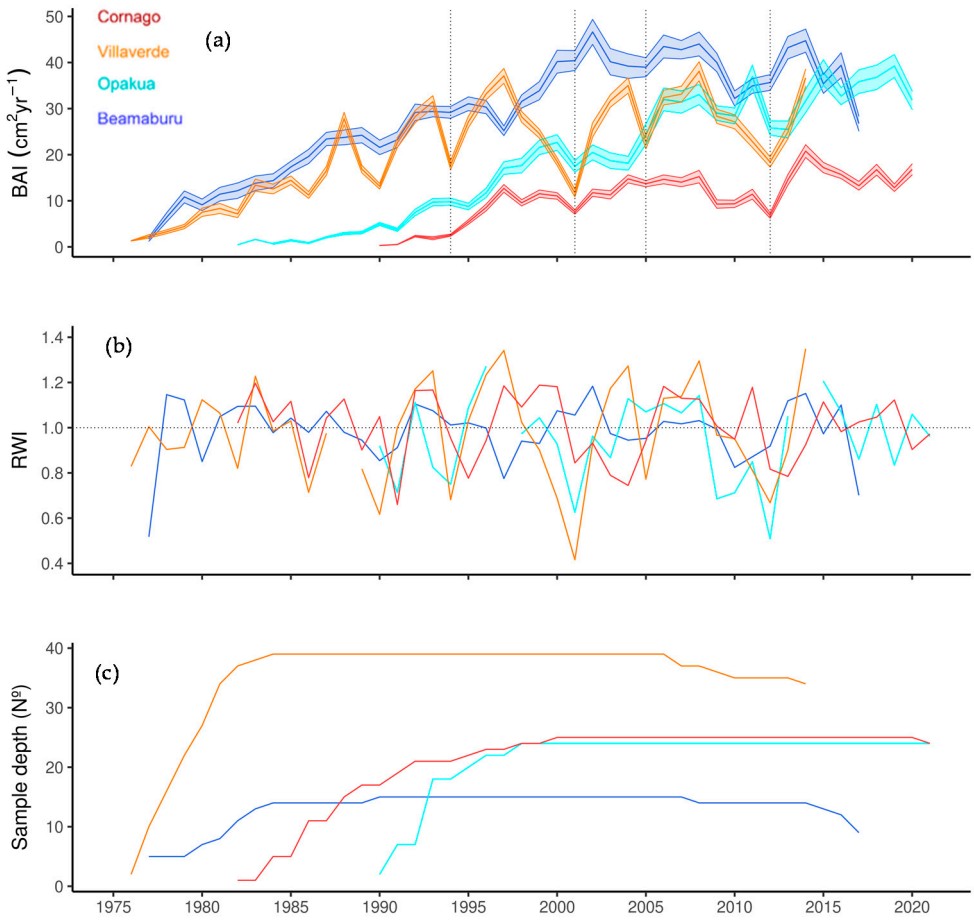

**Figure 2.** Basal area increment ((**a**), BAI) and ring width index ((**b**), RWI) for Douglas fir in the four studied sites. For BAI, solid lines represent mean values across individuals and shaded areas are the 95% confidence intervals for the means. The lowest plot (**c**) shows the sample depth (number of cores). In the uppermost plot, the dotted vertical lines show dry years (1994, 2001, 2005 and 2012).

### 3.2. Relationships between Tree Growth and Climate

The relationship between climate and growth pointed to the stronger sensitivity of Douglas fir growth to drought in the Cornago driest site (Figure 3). Douglas fir growth in Cornago was strongly dependent on the 3-month SPEI of July and August. High temperatures during spring favoured the growth of Douglas fir in Cornago and Villaverde, whereas high temperatures during August reduced the growth in all sites excluding Opakua.

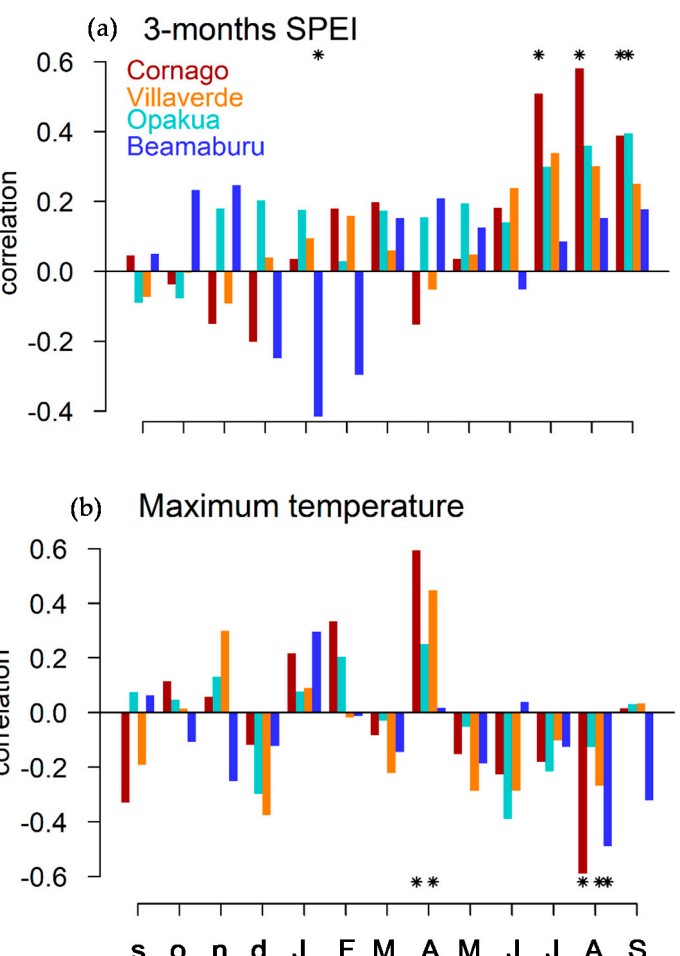

**Figure 3.** Relationships (Pearson correlations) between Douglas fir growth and climate. The relationship between ring-width indices of Douglas fir 2014 with 3-month-long SPEI (**a**) and the mean maximum temperature (**b**) is represented with different colors for each site. Asterisks (*) indicate significant ($p < 0.05$) correlations for each site (colors) in that month (lowercase letters for the previous year and uppercase letter for the current year).

At the tree level, the selected model among those proposed included effects of tree age trend, SPEI and Tmax as well as their interaction with the site (Table S1). Thus, the selected model explained a large proportion of the variation in Douglas fir growth ($R^2c = 0.712$; $R^2m = 0.500$) and confirmed that Douglas fir radial growth dependency on temperature and drought varied between sites (Table 3). These results are contingent on the existence of differences in growth conditions across sites. For example, if we project the BAI of trees using average age values across sites, then the Cornago site displays the lowest growth values and the Opakua site the highest values (Figure 4).

**Table 3.** Summary of the linear mixed-effect models proposed to study growth trajectories and response to climate of Douglas fir. Variables' abbreviations: df, degrees of freedom; SPEI, 3-month August SPEI; Tmax, August maximum temperature. Ages are log-transformed data.

| Variable | Estimate | Std. Error | df | *t* Value | *p* |
|---|---|---|---|---|---|
| Intercept | 34.276 | 1.726 | 74.631 | 19.860 | 0.000 |
| Cornago | −19.584 | 2.684 | 83.189 | −7.297 | 0.000 |
| Opakua | −9.413 | 2.402 | 64.290 | −3.919 | 0.000 |
| Villaverde | −8.287 | 2.117 | 59.871 | −3.914 | 0.000 |

**Table 3.** *Cont.*

| Variable | Estimate | Std. Error | df | *t* Value | *p* |
|---|---|---|---|---|---|
| SPEI | −0.487 | 0.891 | 44.433 | −0.546 | 0.588 |
| Tmax | −2.481 | 0.871 | 41.934 | −2.847 | 0.007 |
| Age | 5.740 | 0.919 | 51.780 | 6.243 | 0.000 |
| Cornago: age | −0.865 | 0.875 | 1387.561 | −0.988 | 0.323 |
| Opakua: age | 4.647 | 0.781 | 1399.288 | 5.948 | 0.000 |
| Villaverde: age | −3.607 | 0.695 | 1388.601 | −5.189 | 0.000 |
| Cornago: SPEI | 1.625 | 0.774 | 1378.864 | 2.099 | 0.036 |
| Opakua: SPEI | 2.948 | 0.734 | 1366.048 | 4.018 | 0.000 |
| Villaverde:SPEI | 2.576 | 0.649 | 1367.387 | 3.971 | 0.000 |
| Cornago: Tmax | 1.051 | 0.729 | 1375.562 | 1.440 | 0.150 |
| Opakua: Tmax | 2.566 | 0.708 | 1362.926 | 3.626 | 0.000 |
| Villaverde: Tmax | 1.058 | 0.629 | 1375.644 | 1.683 | 0.093 |

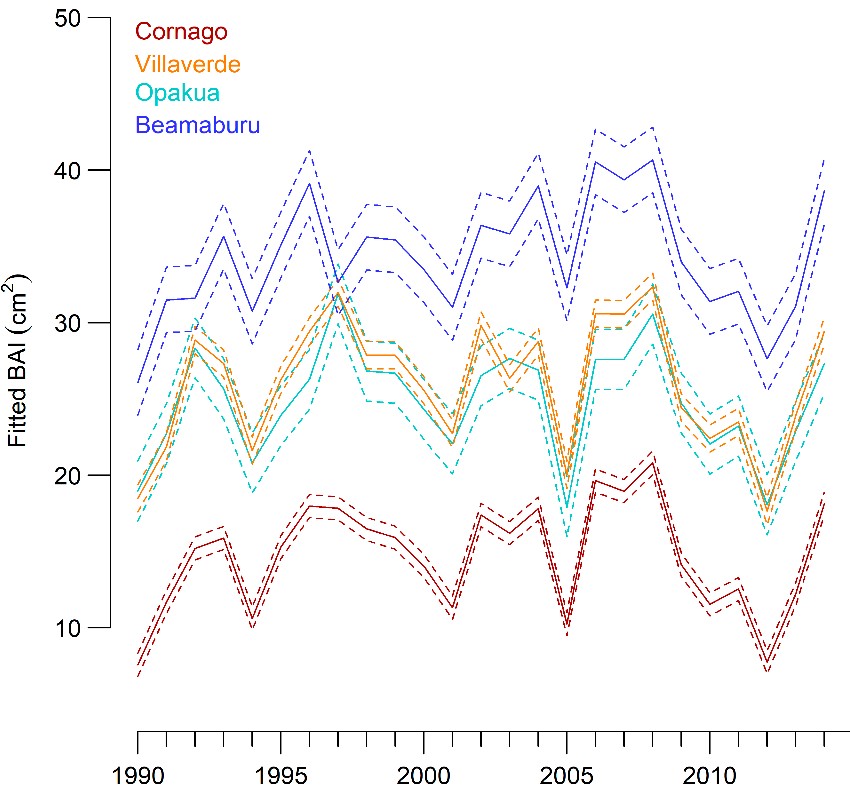

**Figure 4.** Expected growth (BAI, basal area increment) of Douglas fir trees for the period 1990–2014 if all sites are hypothetically forced to have the same age (i.e., standardized mean age values are 0). Continuous lines are means and dashed lines are 95% confidence intervals.

*3.3. Relationships between Wood Density and Climate*

The Douglas fir wood density in the Beamaburu site has a slightly positive trend as trees aged and varied between years (Figure 5). For instance, the wood density residuals showed low values in 1996 and 2010 (Figure S3).

The relationship between wood density, mean maximum temperature and SPEI was, in general, low (Figure 6). We found a positive relationship between prior winter (December and January) SPEI and MND and a negative relationship between MND and April SPEI. The relationship between MXD and SPEI was negative from June to August. By contrast, the relationship between RGD and maximum temperature were positive. Significant positive relationships were observed between MND and February and March temperatures (also for RGD), May temperature and RGD and June temperature and MXD (Figure 6).

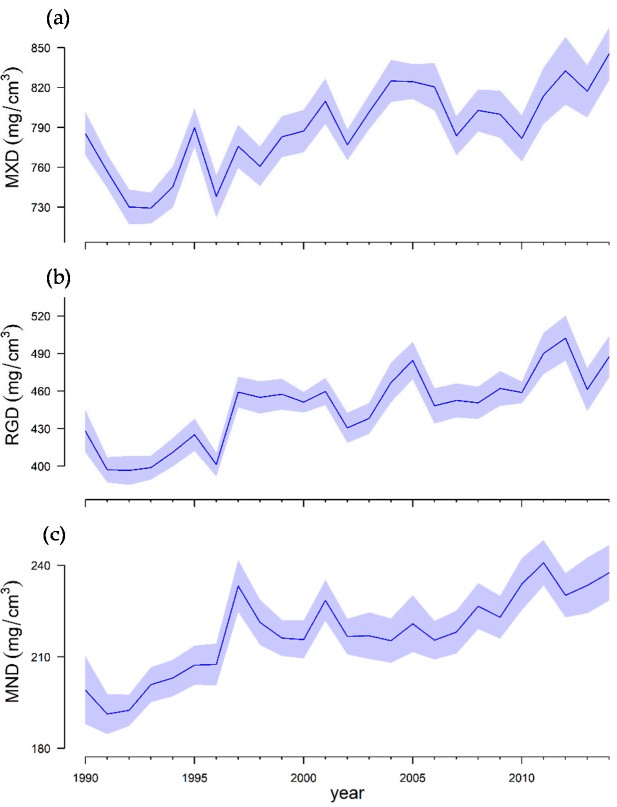

**Figure 5.** Maximum ((**a**), MXD), mean ((**b**), RDG) and minimum ((**c**), MND) wood density of Douglas fir in the Beamaburu site (1990–2014). The figure shows the mean and the 95% confidence intervals across individuals.

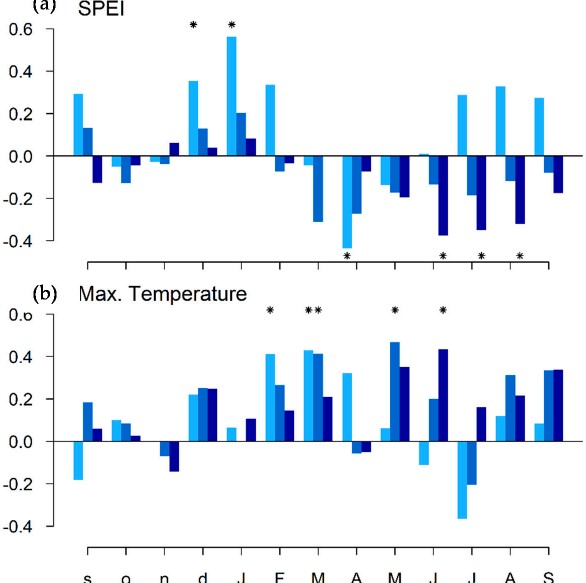

**Figure 6.** Relationship between Douglas fir wood density (detrended indices) and (**a**) monthly SPEI values and (**b**) mean maximum temperatures in the Beamaburu site. The relationships between minimum (MND, light blue bars), ring (RDG, medium blue bars) and maximum (MXD, dark blue bars) wood density are shown for September in the previous year (months abbreviated by lowercase letters) to September in the year of tree-ring formation (months abbreviated by uppercase letters). Significant relationships ($p < 0.05$) are shown with asterisks (*) for each variable.

### 3.4. Climatic Limitations of Tree Growth

According to the VS model, the growth responses to climate differed among sites. Cornago and Villaverde were the sites with the best fitted simulations (Table S2, Figure S4). These were also the two sites where ring-width indices responded more to climate, particularly to reduced soil moisture in late summer, while Beamaburu showed the lowest resposivennes (Figure 7). In Opakua, growth rates also decreased when late-summer soil moisture decreased.

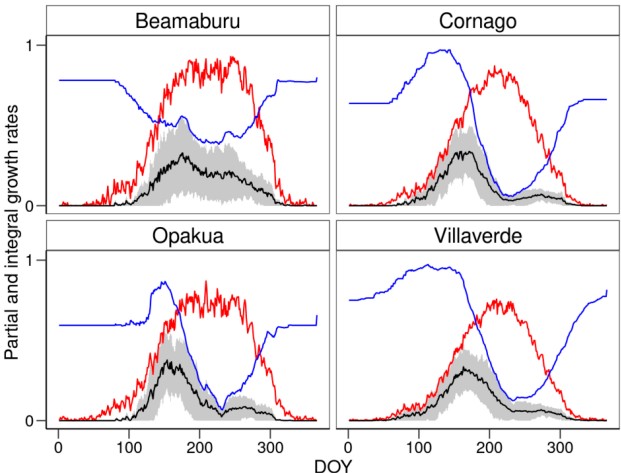

**Figure 7.** Simulated partial growth rates due to temperature (GrT, red lines) and soil moisture limitations (GrM, blue lines) and integral growth rates (Gr, black lines) averaged during the simulation period for each site. The grey lines show the standard deviations of the integral growth rates. DOY is the day of year.

Based on the VS model simulations, low soil moisture was a very important constraint regarding Douglas fir growth during warm–dry decades, such as the 2000s in the Villaverde and Cornago warm–dry sites (Figure 8).

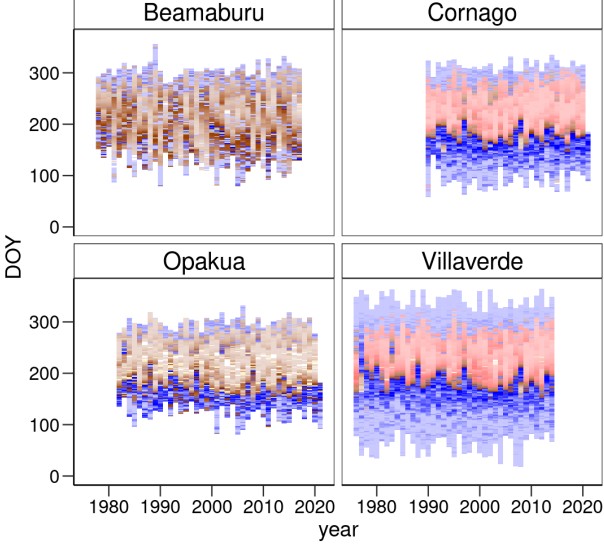

**Figure 8.** Climatic limitations of daily simulated partial growth rates (Gr) throughout the study periods in each site. When low temperature is the main limiting factor of growth (GrT < GrM), the colors of symbols are blue; if the main limiting factor is low soil moisture (GrT > GrM), then symbols are red; when soil moisture limits growth, symbols are brown; the white symbols show optimal climatic conditions (GrT = GrM = 1). DOY is the day of the year.

## 4. Discussion

Here, we show that drought constrains Douglas fir growth in planted forests subjected to seasonally dry climate conditions. In line with our first hypothesis, we found that Douglas fir grew less and suffered more from drought in the Cornago driest site, probably due to the dry conditions prevailing there. These results contrast somewhat with what was found in southern Italy, where no response of Douglas fir to summer drought was found [21]. Such a discrepancy may be explained by the wetter conditions of Italian sites. However, our results concur with their findings on the strong sensitivity of Douglas fir growth to summer maximum temperatures, which probably enhance the atmospheric drought by increasing the vapor pressure deficit constraining Douglas fir growth. The results partially confirmed our second hypothesis as we found that growth of Douglas fir was lower in the dry Cornago site than elsewhere. However, no differences in growth were found between Villaverde and the two wettest sites (Beamaburu and Opakua). These results suggest that Douglas fir has strong potential as a timber species across Europe if drought is not too strong [16]. We also found that wood density fluctuations varied between years as a function of climatic conditions. The negative association between spring SPEI and minimum wood density suggests that Douglas fir growth responds to drought by reducing the earlywood tracheids lumen area, even in wet sites, such as Beamaburu. Finally, the growth model showed that Douglas fir growth was constrained by low soil moisture in late summer in all the sites except the wettest Beamaburu site.

Our results show that growth varied between the sites studied, with lower growth rates as drought increased (Table 2, Figure 2). In fact, if we force all the sites to have similar climate conditions and forecast the expected growth for each site, it becomes clear that the growth of Douglas fir in the driest Cornago site becomes disproportionately smaller than in the rest of the sites (Figure 4). This means that the growth in this site is constrained by warm summer temperatures and low soil water availability, which limit the capacity of Douglas fir to grow (Figure 3), as found in previous studies [21,24] and in agreement with the VS model results (Figure 7).

The planted forests studied are relatively young (Table 2, Figure 2), and this has implications when comparing these results with those found in other studies. For example, the Douglas fir forests studied in southern Italy showed average ages around 65–70 years [21]. Additionally, Ref. [24] showed an increase in Douglas fir growth sensitivity to drought in the Czech Republic using trees at least 50 years old and spanning the period 1965–2015. An enhanced sensitivity of Douglas fir to drought was found studying uneven-aged forests in Austria and Germany [55]. In any case, these studied stands are far from the age ranges that Douglas fir reach in their native ranges [16] and the strong potential that this species has for growing vigorously for several centuries [56].

Both in its native range and in several plantations across Europe and other continents, the results suggest that Douglas fir growth is drought-sensitive. Our results align with these findings, and this is consistent across sites if we look at the severe growth reductions imposed by the 2005 and 2012 droughts (Figures 2 and 4). It is well known that the 2005 drought affected the resilience capacity of the most common tree species across Spain [57]. The effects of the 2012 drought were less consistent across Spain, but it caused strong growth reductions in some tree species [58]. We found some impact of the 2003 central European heatwave on the growth of this species in Cornago and the Villaverde dry site but not in the other two sites even though the wet Beamaburu site is located in a clearly temperate region. This suggests that the studied plantations are mainly influenced by the general climate patterns in Spain and partially decoupled from more central European extreme events. Thus, these planted forests somehow represent natural experiments of how Douglas fir will respond to mainly Mediterranean climate conditions, which has important implications given the commercial importance of this timber species.

An important aspect to consider when dealing with tree species is their capacity for growth and the wood characteristics, such as density, that determine the capacity to uptake and fix carbon in woody tissues [26]. Here, we show that growth of Douglas fir is lower in

the more arid Cornago site than in the other sites. The values of tree-ring width found in this study are, in general, lower than those found in other countries of southern Europe, such as Portugal [59]. However, the values found in our study were larger than those reported in Ref. [24] for the Czech Republic. Nevertheless, these comparisons need to be conducted with caution given the abovementioned differences in tree age or management histories between sites. This reflects that Douglas fir has been planted in sites with a wide variation in climate and soil conditions across the continent [19].

Warmer spring–summer conditions probably increase the minimum wood density through formation of dense earlywood, as has been observed in Mediterranean conifers from drier sites [36,37]. Such anatomical wood features deserve further attention when studying the responses of timber species to climate warming. Here, we have studied intra-ring wood density fluctuations and showed that drought leads to formation of denser earlywood but at the cost of reducing the ring width, which is also constrained by late-summer drought stress manifested as reduced soil moisture. These two wood characteristics, density and total growth (ring width), affect the capacity of trees to fix carbon in woody tissues, so they should be systematically assessed in similar studies comparing planted tree species across climate gradients [60]. Since Douglas fir is not native, different mixtures with coexisting native tree species could be tested using growth, wood anatomical characteristics (e.g., lumen area) or wood density as proxies of drought tolerance [61].

Two caveats on our findings are the lack of information regarding the past management of plantations (we found no indications of past management) and the genetic origin of seeds. Consequently, it is possible that the studied stands were subjected to different management strategies [16], resulting in different potential growth responses to climate. In addition, wood density and hydraulic functioning have shown to vary among Douglas fir provenances [62]. We focused on pure Douglas fir stands to avoid mixing effects [16], but we do not consider other important aspects, such as intra-specific competition. These aspects deserve further attention in future research.

## 5. Conclusions

Douglas fir is being promoted as a timber species with strong potential to mitigate climate change impacts in Europe. Its capacity to tolerate relatively dry conditions and grow fast have been suggested as reasons to promote its usage [4]. However, use of non-native species is not absent of controversy due to their potential impact on multiple ecosystem services provided by natural forests [12]. Doubts also exist on the actual capacity of this species to tolerate drought as recent studies suggest that this species is more vulnerable to drought and heat stress than previously thought in Europe [17]. Here, we show that, in seasonally dry climates with a marked late summer drought, the capacity of Douglas fir to grow is markedly reduced. Plantations in dry sites, such as the Cornago study site, constitute useful, natural experiments that can facilitate study of how Douglas fir can respond to future warmer droughts. Thus, this "sentinel plantation", as well as other similar ones located in rear-edge, southern xeric habitats, should be monitored in order to orient future decision-making in the European forestry sector.

**Supplementary Materials:** The following supporting information can be downloaded at: https://www.mdpi.com/article/10.3390/f13111796/s1, Table S1: Potential linear mixed-effect models proposed to study growth trajectories and response to climate of Douglas fir; Table S2: Statistics of the fits of the VS model; Figure S1: Precipitation and temperature patterns in the four studied sites; Figure S2: trends in annual climate data; Figure S3: Residual variation in the intra-ring density fluctuations; Figure S4: Simulated and observed series of ring-width indices.

**Author Contributions:** Conceptualization, A.G., C.V. and J.J.C.; methodology, A.G., C.V., M.V., A.C. and J.J.C.; software, A.G., C.V. and J.J.C.; validation, C.V. and J.J.C.; formal analysis and investigation, A.G., C.V. and J.J.C.; data curation, C.V. and J.J.C.; writing—original draft preparation, review and editing, A.G., C.V., M.V., A.C. and J.J.C.; visualization, A.G., C.V. and J.J.C.; funding acquisition, A.G., A.C. and J.J.C. All authors have read and agreed to the published version of the manuscript.

**Funding:** This research was funded by: (i) a "Ramón y Cajal" postdoctoral Program of the SPANISH MINISTRY OF SCIENCE, INNOVATION AND UNIVERSITIES under Grant RyC2020-030647-I (A.G.) and (ii) the SPANISH MINISTRY OF SCIENCE, INNOVATION AND UNIVERSITIES, grant number RTI2018-096884-B-C31. A.G. received funding from PIE-20223AT003 of CSIC.

**Data Availability Statement:** Data are available on request from the corresponding author.

**Acknowledgments:** We thank several colleagues for their help during field sampling.

**Conflicts of Interest:** The author declares no conflict of interest.

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
