# Peer review of "Douglas Fir Growth Is Constrained by Drought: Delineating the Climatic Limits of Timber Species under Seasonally Dry Conditions"

_forests, doi:10.3390/f13111796_

Round 1

Reviewer 1 Report

Using dendroclimatological technology and the VS model, this study investigated tree radial growth in response to climate variables in four Douglas fir plantations with contrasting climate regimes in northern Spain. The authors found that radiagrowth of Douglas fir is mostly influenced by climate conditions of the summer and early autumn and the influence is especially pronounced in drier sites. Results of the study based on multiple methods are solid and can inform species selection in afforestation under increasingly hotter and drier climate conditions in EuropeI have the following suggestions for the improvement of this manuscript.

1. In the Introduction section, the authors need to point out where is the knowledge gap and how the current study can fill the gap. Meanwhile, there are too many paragraphs in this section and some of them are recommended to be merged, for instance, paragraph 3, 4 and 5 can be combined into one paragraph.

2. It may be unsuitable to directly relate raw data of wood density with climate variables, as the measurements contain information related to tree age. It is recommended that the metrics of tree-ring density to be detrended before conducting correlation analyses with climate factors, just like that for tree-ring width. 

3. Why the period for correlation analysis between tree-ring metrics and climate variables was determined for the period 1990-2014? In Dendroclimatology, tree-ring indices during reliable period (usually determined by the criterion of EPS>0.85) are used to correlate with climate variables. In the current study, although time series of many tree-ring cores may cover the 1990-2014 period, it does not mean that tree-ring indices during this period are reliable. Need to clarify.

4. The discussion section needs to be substantially improved. In the current version, many parts are onlrepeating the results (i.e. lines 382-385, 408-412, 416-417) and the connections between them are a bit loose. 

Minor comments:

Line 168: The monthly total precipitation data from CRU database do not seem to be employed in this study.

Line 282: It is recommended to add a curve to show the change of sample size over time in Figure 2. 

Line 452: Change than to that.

Author Response

Comments and Suggestions for Authors

Using dendroclimatological technology and the VS model, this study investigated tree radial growth in response to climate variables in four Douglas fir plantations with contrasting climate regimes in northern Spain. The authors found that radial growth of Douglas fir is mostly influenced by climate conditions of the summer and early autumn and the influence is especially pronounced in drier sites. Results of the study based on multiple methods are solid and can inform species selection in afforestation under increasingly hotter and drier climate conditions in Europe.

> We thank the reviewer for finding merit in our study.

I have the following suggestions for the improvement of this manuscript.

  1. In the Introduction section, the authors need to point out where is the knowledge gap and how the current study can fill the gap. Meanwhile, there are too many paragraphs in this section and some of them are recommended to be merged, for instance, paragraph 3, 4 and 5 can be combined into one paragraph.

> We rephrased and revised the Introduction as suggested.

  1. It may be unsuitable to directly relate raw data of wood density with climate variables, as the measurements contain information related to tree age. It is recommended that the metrics of tree-ring density to be detrended before conducting correlation analyses with climate factors, just like that for tree-ring width.

> We accounted for tree age effects in the models and used detrended (age-corrected) density data in the analyses. This is indicated in the revised ms.

  1. Why the period for correlation analysis between tree-ring metrics and climate variables was determined for the period 1990-2014? In Dendroclimatology, tree-ring indices during reliable period (usually determined by the criterion of EPS>0.85) are used to correlate with climate variables. In the current study, although time series of many tree-ring cores may cover the 1990-2014 period, it does not mean that tree-ring indices during this period are reliable. Need to clarify.

> We considered this period because it was the best replicated and we got EPS > 0.85 in all sites. We clarified this issue in the revised ms.

  1. The discussion section needs to be substantially improved. In the current version, many parts are only repeating the results (i.e. lines 382-385, 408-412, 416-417) and the connections between them are a bit loose.

> We removed or rephrased redundant sentences.

Minor comments:

Line 168: The monthly total precipitation data from CRU database do not seem to be employed in this study.

> Yes, we used them to calculate climate-growth relationships (Figure 3).

Line 282: It is recommended to add a curve to show the change of sample size over time in Figure 2.

> Done, we added it.

Line 452: Change “than” to “that”.

> Done, we changed it.

Reviewer 2 Report

It is a good idea to evaluate the growth response to climate variables and drought of timber species (Douglas Fir) in Europe, which will help the governments and commercial companies to estimate how the non-native tree species will thrive under more arid conditions. In this study, some evidence from the tree-ring width and wood density have shown that Douglas fir is drought tolerance and is constrained by warm and dry conditions during summer and early autumn, particularly in the driest study site. However, there are still some problems unresolved in the manuscript, as well as the writing and figures should be improved before it can be considered to publish.

Main concerns:

1.       There is an interesting result shown in the Figure 2: the BAI in the most moisture site (Beamaburu) indicates declining trend after 2000, while the BAI keep increasing in the two drought sites (Cornago and Villaverde). Why? 

2.       Is the rising trend of wood density shown in Figure 5 related with the tree-age?

3.       I do not agree with that “no clear evidence of the impact of the 2003 central European heatwave on the growth of this species… (Line398-402)”. If you keep under observation on the Figure 8, you will find the low soil moisture effects on tree-growth in 2003 are obvious in Cornago and Villaverde, and the soil moisture limiting happened even earlier than other years. More analysis about the heat effects on tree growth should be conducted, especially the title of manuscript highlight the “drought and heat” effects.

4.       From the Figure 2, there is no obvious evidence (tree growth reductions) to support the drought effects in 2005 and 2012.

5.       How to get the results in Figure 4? What is the purpose of Figure 4? It should be stated more clearly.

6.       Line 235, what is the purpose of comparing mean tree-ring values between sites (different climatic and environmental conditions)? Sounds meaningless.

Revise suggestions:

7.       The research hypothesis (1) and (2) communicate the same idea. Also, I suggest providing the study aims and the framework to highlight the core focus of this paper.

8.       Line 132-136, it will be better to list the basic information in a table.

9.       The Materials and Methods section, 1) the subtitles are too much; 2) the subtitle of 2.5 is improper: the main point is about the detrending method and how to estimate the tree-ring chronology, while no climate-growth relationships were stated in this paragraph. 3) Line 214-222, this paragraph should not be included in the “wood density data”.

10.   Line 374-376, there is no evidence to support “ Douglas fir has a strong potential to replace other native species such as European beech” in this paper, as only one tree species was analyzed.

11.   The organization of the Discussion is not clear and logical, especially for the Line 416-435.

12.   There is semantic contradiction in the Line 446-449.

Figure problems:

13.   Figure 1, the locations of the four studied sites should be plotted in the map to help the readers to understand the study area. Also, the climate changes for the four sites should be provided (not just the intra-annual variation in Figure S1).

14.   Figure 8, the meanings of the green and black symbols should be stated more clearly.

15.   The picture quality is not high. For example, the figure 2 (not clear and the X-axis is not uniform), figure 5 (not clear), figure 6 (there is no Y-axis for plot b) and no lables), figure S1 (the lines overlapping on the bars), figure S3 (not clear).

Author Response

Comments and Suggestions for Authors

It is a good idea to evaluate the growth response to climate variables and drought of timber species (Douglas Fir) in Europe, which will help the governments and commercial companies to estimate how the non-native tree species will thrive under more arid conditions. In this study, some evidence from the tree-ring width and wood density have shown that Douglas fir is drought tolerance and is constrained by warm and dry conditions during summer and early autumn, particularly in the driest study site. However, there are still some problems unresolved in the manuscript, as well as the writing and figures should be improved before it can be considered to publish.

> We thank the reviewer for finding merit in our study. We have improved the writing and figures following his/her recommendations.

Main concerns:

  1. There is an interesting result shown in the Figure 2: the BAI in the most moisture site (Beamaburu) indicates declining trend after 2000, while the BAI keep increasing in the two drought sites (Cornago and Villaverde). Why?
  2. Is the rising trend of wood density shown in Figure 5 related with the tree-age?

> The post-2000 BAI trend in Beamaburu was not significant.

> No, since we studied plantations where trees have the same age. In Figure 5, this could be the case (and this is now mentioned in the revised ms.) but we removed age trends when calculating climate-density correlations.

  1. I do not agree with that “no clear evidence of the impact of the 2003 central European heatwave on the growth of this species… (Line398-402)”. If you keep under observation on the Figure 8, you will find the low soil moisture effects on tree-growth in 2003 are obvious in Cornago and Villaverde, and the soil moisture limiting happened even earlier than other years. More analysis about the heat effects on tree growth should be conducted, especially the title of manuscript highlight the “drought and heat” effects.

> We agree and rephrased the sentence. We think the analyses of high temperatures on growth are well covered by Figs. 3 and 6 (note also that the SPEI depends on PET which is tightly related to maximum temperatures).

  1. From the Figure 2, there is no obvious evidence (tree growth reductions) to support the drought effects in 2005 and 2012.
  2. How to get the results in Figure 4? What is the purpose of Figure 4? It should be stated more clearly.

> We think Fig. 2a shows growth reductions in 2005 and 2012 in Villaverde and Cornago sites.

> We think that Figure 4 is informative enough because it shows age-corrected growth trends in the four study sites. These are fitted values according to the mixed-effect model for individuals with equal age across sites. Thus, it allows to differentiate growth and response to SPEI across sites. The figure clearly shows that Douglas fir growth is lower in Cornago and that all sites show growth reductions in 2005 and 2012 dry years. We have modified the results section and figure legend accordingly.

  1. Line 235, what is the purpose of comparing mean tree-ring values between sites (different climatic and environmental conditions)? Sounds meaningless.

> We compared growth rates between sites to detect differences due to climatic stress.

Revise suggestions:

  1. The research hypothesis (1) and (2) communicate the same idea. Also, I suggest providing the study aims and the framework to highlight the core focus of this paper.

> We rephrased and joined the hypotheses and provided a study framework.

  1. Line 132-136, it will be better to list the basic information in a table.

> Done. We created a new Table, but we added the main features and not dendrometric variables because of lack of space.

  1. The Materials and Methods section, 1) the subtitles are too much; 2) the subtitle of 2.5 is improper: the main point is about the detrending method and how to estimate the tree-ring chronology, while no climate-growth relationships were stated in this paragraph. 3) Line 214-222, this paragraph should not be included in the “wood density data”.

> We shortened and rephrased some of the subtitles.

> We shortened some of the subtitles.

> We moved the mentioned paragraph to a new section.

  1. Line 374-376, there is no evidence to support “ Douglas fir has a strong potential to replace other native species such as European beech” in this paper, as only one tree species was analyzed.

> We removed the sentence.

  1. The organization of the Discussion is not clear and logical, especially for the Line 416-435.

> We reorganized and rephrased the Discussion.

  1. There is semantic contradiction in the Line 446-449.

> We rephrased to avoid contradictory arguments.

Figure problems:

  1. Figure 1, the locations of the four studied sites should be plotted in the map to help the readers to understand the study area.

> Following your recommendation, we added a map showing the location of the four study sites in N Spain and Europe. See new Figure 1.

Also, the climate changes for the four sites should be provided (not just the intra-annual variation in Figure S1).

> We showed them in the new Fig. S2.

  1. Figure 8, the meanings of the green and black symbols should be stated more clearly.

> Done, we changed and corrected it.

  1. The picture quality is not high. For example, the figure 2 (not clear and the X-axis is not uniform), figure 5 (not clear), figure 6 (there is no Y-axis for plot b) and no lables), figure S1 (the lines overlapping on the bars), figure S3 (not clear).

> We corrected the figures following your recommendations. We improved the quality of Figures 2 and 5 and added the Y-axis for panel “b” in Figure 6. We replotted Fig S1 without overlapping lines but the result was worse and did not reflect the dry period as in typical climate diagrams. So, we kept the original Fig. S1.

Round 2

Reviewer 2 Report

Thanks for your revision. I am satisfied with the changes you have made. I think the current version is appropriate for publication in Forests.